# SARS-CoV-2 and Parkinson’s Disease: A Review of Where We Are Now

**DOI:** 10.3390/biomedicines11092524

**Published:** 2023-09-13

**Authors:** Iro Boura, Mubasher A. Qamar, Francesco Daddoveri, Valentina Leta, Karolina Poplawska-Domaszewicz, Cristian Falup-Pecurariu, K. Ray Chaudhuri

**Affiliations:** 1Parkinson’s Foundation Centre of Excellence, King’s College Hospital, Denmark Hill, London SE5 9RS, UK; iro.boura@kcl.ac.uk (I.B.);; 2Basic and Clinical Neuroscience, The Maurice Wohl Clinical Neuroscience Institute, Institute of Psychiatry, Psychology and Neuroscience, King’s College London, 5 Cutcombe Road, London SE5 9RX, UK; 3Medical School, University of Crete, Heraklion, 71003 Iraklion, Greece; 4Department of Translational Research and of New Surgical and Medical Technologies, University of Pisa, Via Savi 10, 56126 Pisa, Italy; 5Parkinson and Movement Disorders Unit, Department of Clinical Neuroscience, Fondazione, IRCCS Istituto Neurologico Carlo Besta, 20133 Milan, Italy; 6Department of Neurology, Poznan University of Medical Sciences, 60-355 Poznan, Poland; 7Department of Neurology, County Clinic Hospital, Faculty of Medicine, Transilvania University Brasov, 500019 Brasov, Romania

**Keywords:** SARS-CoV-2, COVID-19, parkinsonism, Parkinson’s disease, pandemic, vaccination, pathophysiology, infection

## Abstract

Severe acute respiratory syndrome coronavirus-2 (SARS-CoV-2), which causes coronavirus disease 2019 (COVID-19), has been discussed in the context of Parkinson’s disease (PD) over the last three years. Now that we are entering the long-term phase of this pandemic, we are intrigued to look back and see how and why the community of patients with PD was impacted and what knowledge we have collected so far. The relationship between COVID-19 and PD is likely multifactorial in nature. Similar to other systemic infections, a probable worsening of PD symptoms secondary to COVID-19, either transient or persistent (long COVID), has been demonstrated, while the COVID-19-related mortality of PD patients may be increased compared to the general population. These observations could be attributed to direct or indirect damage from SARS-CoV-2 in the central nervous system (CNS) or could result from general infection-related parameters (e.g., hospitalization or drugs) and the sequelae of the COVID-19 pandemic (e.g., quarantine). A growing number of cases of new-onset parkinsonism or PD following SARS-CoV-2 infection have been reported, either closely (post-infectious) or remotely (para-infectious) after a COVID-19 diagnosis, although such a link remains hypothetical. The pathophysiological substrate of these phenomena remains elusive; however, research studies, particularly pathology studies, have suggested various COVID-19-induced degenerative changes with potential associations with PD/parkinsonism. We review the literature to date for answers considering the relationship between SARS-CoV-2 infection and PD/parkinsonism, examining pathophysiology, clinical manifestations, vaccination, and future directions.

## 1. Introduction

Severe acute respiratory syndrome coronavirus-2 (SARS-CoV-2), which causes coronavirus disease 2019 (COVID-19), has swept the world and profoundly affected global health over the past three years, causing widespread illness and death. Although COVID-19 primarily affects the respiratory system, it has been associated with a range of neurological symptoms, including confusion, memory problems and seizures. Furthermore, COVID-19 may contribute to the development of parkinsonian features, including tremor, rigidity and bradykinesia [1]. Many of these cases of parkinsonism were initially linked to COVID-19-induced encephalopathy; however, an increasing number of cases of post-infectious parkinsonism without encephalopathy are being reported [2,3]. Moreover, as discussed below, a clear impact of COVID-19 on people with Parkinson’s disease (PD) has been shown [4].

PD is an age-related and largely heterogeneous neurodegenerative condition, which is primarily linked to the death of dopaminergic neurons in the substantia nigra pars compacta (SNpc), leading to a constellation of motor (bradykinesia, tremor, rigidity and gait/balance deterioration) and non-motor symptoms (NMSs) such as cognitive changes, mood impairment, sleep problems and autonomic dysfunction [5]. The widespread accumulation of intraneuronal cytoplasmic inclusions of α-synuclein (Lewy bodies) is considered a pathological trademark of PD [6]. At the time of diagnosis, an essential number of dopaminergic neurons has already been lost, and neurodegeneration has progressed in the central nervous system (CNS) [7]. Although the pathogenesis of PD remains elusive, it is believed to be a result of both genetic and environmental risk factors, with numerous theories having been published to date. Infectious agents, including viruses, have long been presumed to play a role in the pathogenesis of PD, as discussed below.

Numerous viral infections have been associated with the development of either parkinsonism or PD, such as Epstein-Barr Virus (EBV) and Japanese Encephalitis Virus (JEV) [3,8,9]. Indicatively, individuals with EBV encephalitis presented with new-onset parkinsonism, while imaging confirmed decreased perfusion at the basal ganglia, thus suggesting a direct brain invasion or an injury mediated by post-infectious autoantibodies [10,11,12]. On the other hand, the incidence of PD was reported to be three times higher in human immunodeficiency virus (HIV)-infected individuals compared to the general population [13]. Other studies have provided further support that the risk of PD increases with the severity of HIV [14,15], suggesting that HIV infection can not only lead to parkinsonism but also to PD specifically. A large cohort study in Korea found an increased incidence of PD among patients with hepatitis C and/or B compared to non-infected individuals [16]. There is growing evidence that different strains of the influenza virus may infiltrate the CNS and be particularly associated with the emergence of parkinsonism or PD [17]. According to a UK-based study, exposure to influenza did not necessarily lead to the development of PD but rather to parkinsonism [18], while a recent Danish study reported that infection with influenza virus was associated with a PD diagnosis after ten years had passed since the initial infection [19].

Viruses are thought to cause inflammation in the brain or trigger autoimmune responses which may potentially affect dopamine-producing neurons, leading to parkinsonism or PD [20,21]. To fully appreciate the potential risks, we need to further expand our understanding of the underlying mechanisms leading to the development of neurodegenerative conditions such as PD. This paper focuses on where we stand with respect to understanding COVID-19 and any potential links to PD and/or parkinsonism.

## 2. Does COVID-19 Increase the Risk of Developing Parkinson’s Disease and/or Parkinsonism?

As already mentioned, infectious agents may be involved in the pathogenesis of PD but also in post- and para-infectious cases of persistent or transient parkinsonism [3]. Numerous theories and hypotheses have linked COVID-19 to the development of new-onset parkinsonian features, with a growing number of relevant cases being reported to date. However, at the moment, significant limitations prevent us from making any safe conclusions about the general population.

### 2.1. Currently Available Data

More specifically, 20 cases of parkinsonism were reported up to the beginning of 2022, including 4 cases with a diagnosis of typical PD [2]. In most patients, parkinsonism presented within the first month of the onset of COVID-19, while the severity of COVID-19 seemed to be a risk factor. Prodromal features of PD were rarely reported in the pre-COVID-19 period. The laboratory tests performed were highly heterogeneous and revealed a variety of mechanisms underlying secondary parkinsonism, including hypoxia, vascular events, extra-pontine osmotic demyelination due to hyperglycemia and acute necrotizing encephalopathy. There were also indications of subjacent inflammation in the CNS (findings of lumbar puncture and a fluorodeoxyglucose (FDG)-based PET scan) and an immune-mediated substrate in selected cases (good response to immunomodulatory/immunosuppressive therapy). Finally, dopaminergic uptake imaging was abnormal in all seven patients it was performed on, while most patients exhibited a good response to dopaminergic treatment.

A case series describing six patients diagnosed with PD at an average of 6 weeks after testing positive for COVID-19 was recently described [22]. No association with the severity of COVID-19 was detected, while half of these patients reported a prior history of rapid eye movement (REM) sleep behavior disorder (RBD). Brain MRI findings were generally unremarkable, while a dopaminergic transport (DaT) scan was performed in four cases, demonstrating a decreased uptake in the striatum unilateral or bilaterally. This latter finding, combined with the pre-existent NMSs, led the authors to conclude that COVID-19 infection might have uncovered an ongoing, sub-clinical neurodegenerative procedure in vulnerable patients.

In line with the above, a recent review of 27 patients reported that new-onset movement disorders, including three cases of parkinsonism, usually appeared approximately two weeks after the beginning of COVID-19 and persisted after an average of 7.5 weeks in 22% of patients [23]. Severe COVID-19 and encephalopathy were identified as potential risk factors.

A case report of parkinsonism and anxiety symptoms appearing three weeks after a mild COVID-19 infection was recently reported [24]. The symptoms did not respond to levodopa therapy but significantly improved after the administration of steroids, suggesting an immune-mediated substrate. Another case report of parkinsonism and akinetic mutism due to osmotic demyelination syndrome after COVID-19 was described in a diabetic female patient [25]. The symptoms responded gradually, but significantly, to dopaminergic therapy within a period of two months. Finally, a case of asymmetric parkinsonism developing three months after severe COVID-19 with concomitant global cognitive impairment and pseudobulbar affect, an abnormal DaT scan and a good response to levodopa revealed a subjacent cerebral autosomal dominant arteriopathy with subcortical infarcts and leukoencephalopathy (CADASIL) [26].

### 2.2. Limitations/Future Directions

Several limitations are acknowledged in interpreting the above results. Published cases of parkinsonism largely lack follow-up information or neurological assessment prior to COVID-19. Moreover, the availability of diagnostic tests performed, including advanced imaging testing, is highly variable among different clinical settings, thus affecting researchers’ conclusions. For example, the performance of an FDG-based PET scan allowed authors to detect autoimmune encephalitis, as abnormal findings in the affected areas of the mesial temporal lobes, basal ganglia and brainstem normalized after administering intravenous immunoglobin therapy [27,28]. The sole performance of MRI would not be useful on these occasions as their findings were, indeed, unremarkable. Notably, the association of the aforementioned cases with COVID-19 is mostly attributed to the time of the onset of PD/parkinsonism, meaning that it occurred relatively closely to the COVID-19 diagnosis. However, all the above patients who underwent a lumbar puncture tested negative in an RT-PCR for SARS-CoV-2 in their cerebrospinal fluid (CSF); thus, acute SARS-CoV-2 infection in the CNS could not be objectively confirmed.

Interestingly, a retrospective study explored the possibility of developing PD within two years following a SARS-CoV-2 infection [29]. The researchers compared two large (with more than two million individuals in each cohort) age-, sex- and smoking-history-matched cohorts of individuals with and without COVID-19, revealing an increased risk of new-onset PD during the first year after contracting SARS-CoV-2, which subsequently subsided. Moreover, a recent meta-analysis found a significant association between COVID-19 and several new-onset neurodegenerative disorders, including PD (HR = 1.44 (1.06–1.95), I^2^ = 86%) [30]. Although these results cannot be ignored, the number of published cases remains quite limited to support a hypothesis of COVID-19 being a potential risk factor for new-onset PD or parkinsonism.

The currently available data, though not conclusive, cannot be treated lightly, especially since neurodegeneration is a chronic and multi-level process. Focused studies with large sample sizes and high-quality methodology (e.g., a prospective design and a long follow-up period) are needed to further clarify any potential links of SARS-CoV-2 with PD/parkinsonism, particularly in the long term. A careful description of the involved clinical entities (e.g., a laboratory confirmation of COVID-19 and a clear distinction between a diagnosis of PD or parkinsonism or separate symptoms, such as tremor) is crucial. Furthermore, in order to reduce bias, the collection of a variety of data considering either COVID-19 (e.g., severity, medications used and concomitant encephalopathy) or PD/parkinsonism (e.g., severity, prodromal features, accompanying NMS, thorough imaging and laboratory testing and response to therapy) is essential. Other factors which may further increase the risk of the emergence of PD after a SARS-CoV-2 infection, including age, frailty, past CNS infections, head trauma or other comorbidities, should be examined as well. The impact of vaccination or multiple SARS-CoV-2 infections is currently lacking and should also be considered.

## 3. Is COVID-19 More Common among Parkinson’s Disease Patients?

No robust evidence is currently available that PD patients are more susceptible at developing COVID-19 compared to the general population [31].

### 3.1. Currently Available Data

A recent systematic review and meta-analysis found that PD patients were at a higher risk of contracting SARS-CoV-2 (OR = 1.65 (1.34–2.04), especially if they were cognitively impaired [32]. An observational, retrospective, multicenter study (*n* = 552) showed that the incidence of COVID-19 was significantly higher among PD patients who were younger than 50 and had shorter durations of disease or had less-advanced PD [33]. Three systematic reviews and meta-analyses also explored this hypothesis, highlighting obesity, subjacent pulmonary disease, diabetes mellitus and immune compromise as potential risk factors for the PD population, while another systematic review found outstanding differences in the prevalence of COVID-19 depending on the geographical location of the assessed PD cohort [34,35,36,37]. Interestingly, vitamin D supplementation was noted as a protective factor for COVID-19 [34]. Finally, Afraie and colleagues found no difference in the prevalence of COVID-19 between PD and non-PD individuals in their recent review and meta-analysis of published studies [38].

### 3.2. Limitations/Future Directions

Researchers have commented on the considerable heterogeneity and inadequate data control for significant factors in some of the studies included in the pooled analyses [34,35]. Moreover, self-isolation may have biased the estimate of the true risk of PD patients contracting SARS-CoV-2 as they tended to self-isolate more than their non-PD counterparts (31). Whether this pattern was deliberate or not remains unclear. The true frequency of COVID-19 among PD patients’ needs to be further explored in large studies with representative control groups in which cultural/ethnic diversity and local parameters (e.g., urban versus rural settings) will be accounted for [39].

## 4. Does COVID-19 Worsen the Symptoms of Parkinson’s Disease?

PD patients experiencing an aggravation of parkinsonian features during the acute phase of COVID-19, especially if severe, would not come as a surprise. Clinicians are well aware that a subacute worsening of PD symptoms, both motor and NMSs, is likely to occur whenever people with PD undergo a systematic infection, with fever and delirium being particularly aggravating factors [40]. These phenomena are usually transient, although they may also persist in the long term.

### 4.1. Motor Symptoms/Currently Available Data

The currently available data considering PD patients with confirmed SARS-CoV-2 infections are limited and mostly based on small studies and case series (Table 1). According to the large, online case–control study Fox Insight (*n* = 7209), PD patients with COVID-19 self-reported the deterioration of motor symptoms more often compared to those without COVID-19 [41]. The same trend was also detected by a cross-sectional online study (*n* = 568), with PD patients with concomitant COVID-19 self-reporting a worsening of PD-related motor symptoms, particularly motor fluctuations, more often compared to PD patients without COVID-19 [42]. A community-based case–control study of non-hospitalized PD patients (*n* = 48) found that SARS-CoV-2 infection was associated with the deterioration of motor performance, particularly a reduction in the OFF time (referring to the time periods during the day when PD symptoms may not respond as expected to antiparkinsonian medication and patients experience a deterioration of their motor and/or NMSs), which necessitated an increase in dopaminergic therapy in one-third of the patients [43]. Finally, a case series of ten hospitalized PD patients, a case series of eight PD patients and a case report of two PD patients also found an aggravation of motor symptoms during COVID-19, with alterations in dopaminergic therapy considered necessary on some occasions [44,45,46]. However, according to a case–control study (*n* = 740), no significant differences in motor symptoms or NMSs were observed between PD patients with or without COVID-19 [47].

### 4.2. Non-Motor Symptoms/Currently Available Data

Although changes in NMSs were consistently documented among PD patients during the COVID-19 pandemic, most of these reports referred to PD patients overall without discriminating between those infected with SARS-CoV-2 or not (Table 1). New-onset or the aggravation of depressive symptoms and anxiety during COVID-19 were documented in the Fox Insight study, with women being at a higher risk, and in case series describing 10 hospitalized patients [41,44]. Similarly, new-onset or the deterioration of fatigue or sleep problems were reported almost three times and twice as often, respectively, among PD patients with COVID-19 compared to those without COVID-19 in the Fox Insight study [41]. The latter trend was more potent among female participants and individuals who experienced interruptions in their PD-related medical care and/or social activities/exercise. Moreover, in a community-based study of PD patients (*n* = 48), fatigue was reported significantly more frequently among those who were affected by COVID-19 compared to those who were not (58.4% vs. 8.3%, *p* < 0.001) [43], while it was described as a cardinal symptom in the case series of Antonini and colleagues [44].

Cognitive impairment was more consistently reported among PD patients with COVID-19. In a small, multicenter 12-week study (*n* = 27), new-onset cognitive problems appeared in 22% of patients, including “brain fog”, the impairment of concentration and memory deficits [4]. In an online survey of 46 PD patients with COVID-19, almost half reported intellectual impairment [48], while in the aforementioned case series of Antonini et al., cognition was also found to worsen during SARS-CoV-2 infection [44]. Finally, in the community-based study of Cilia et al. comparing PD patients with and without COVID-19, cognitive performance was only marginally affected in the former group [43].

Reports of other NMSs among PD patients with COVID-19 were rather scarce. In a survey of 46 PD patients with COVID-19 at the Columbia University Irving Medical Center, 40% described a worsening of pain and 4% noticed new-onset pain following the infection [48]. Dysautonomia among PD patients was also reported significantly more often among those with COVID-19 compared to those without, particularly orthostatic hypotension, urinary urge incontinence and nocturia [41,43,44].

### 4.3. Limitations/Effect of Quarantine/Future Directions

Most of the above data were retrospectively collected during the pandemic when strict restrictions were imposed on social encounters and access to healthcare and rehabilitation facilities was limited. It may thus be challenging to differentiate between the effects of COVID-19 per se from the impact of the pandemic-associated stress [58] and isolation [59] and also from the potentially poor management of PD symptoms during that phase due to interrupted medical care. On the other hand, two online cross-sectional studies (342 PD patients and 113 non-PD relatives) showed that medical care for PD patients was persistently troublesome, even after quarantine measures were loosened [60]. People with PD reported a continuous aggravation of their symptoms, resulting in lower quality of life for both them and their caregivers, with younger patients and those with longer disease durations being at an increased risk. The re-initiation of physiotherapy after the COVID-19 quarantine was not found to fully reverse the deterioration of motor performance among PD patients who had to interrupt their regular schedule of physiotherapy [61]. Data from well-organized studies with large samples that refer to the time after the quarantine or systematic vaccination are necessary to complement our available knowledge on the effect of SARS-CoV-2 on PD.

## 5. Does COVID-19 Have Long-Term Complications in Parkinson’s Disease?

In the case of persistent or de novo symptoms appearing four weeks or longer after the onset of acute COVID-19, the term “long-COVID” can be used. The reports of long COVID symptoms in PD are limited.

### 5.1. Currently Available Data

According to a case series describing 27 PD patients, the majority of enrolled patients developed long COVID symptoms, including the aggravation of motor symptoms (52%), fatigue (41%), cognitive disturbances (22%) and sleep problems (22%) [62]. Long COVID was not associated with the severity of COVID-19. A recent prospective study of PD patients with a history of COVID-19 (38 with and 20 without persistent post-COVID-19 symptoms) found a statistically significant difference in the equivalent dose of levodopa and motor performance at 6 months after the COVID-19 diagnosis for the former group [63]. A new onset of NMSs was also noted among these patients, including anosmia/hyposmia, sore throat and taste alterations.

### 5.2. Limitations/Future Directions

Due to the limited number of published cases the above data should be received with caution. Since there is no regular follow-up of these patients, the frequency of long COVID symptoms may have been underestimated. Greater vigilance is recommended for treating physicians in order to efficiently approach PD patients with past COVID-19 and accurately detect such symptoms.

## 6. Are There Any Particular Biomarkers for PD Patients with a History of COVID-19?

At present, no biomarkers have been established to efficiently track the progression of PD after a SARS-CoV-2 infection. Clinical biomarkers, including regular recordings of NMSs via standardized methods, would be essential, especially in the context of long COVID. Imaging biomarkers could also be of use as a recent study of 785 UK Biobank participants showed a significantly decreased global brain size in addition to reductions in gray matter thickness and tissue contrast in selected brain regions after COVID-19 [64]. These changes were combined with a decline in the average cognitive status, highlighting the neurodegenerative potential of the virus, although the reversibility of these findings remains unknown. Finally, a recent study of 36 elderly PD patients hospitalized with COVID-19 pneumonia found that low levels of albumin were associated with higher mortality [65].

## 7. Does COVID-19 Increase the Hospitalization Rate or Mortality in Parkinson’s Disease?

The hospitalization rate of PD patients during SARS-CoV-2 infection ranges from 29 to 49%, with Chambergo-Michelot et al. commenting that this rate is independently associated with COVID-19- and PD-related complications (which may be secondary to COVID-19, e.g., motor fluctuations due to vomiting) [34,36,37]. This rate was not found to differ from the hospitalization rate of an age-matched subpopulation [66], while the length of hospitalization was similar between COVID-19 patients with or without PD [35].

Five systematic reviews and meta-analyses explored mortality rates among PD patients with COVID-19, concluding that PD may be associated with increased mortality and poor outcomes [32,35,36,37,67]. Although initial estimations varied greatly due to the small number of patients, the study design and the excessive heterogeneity of the collected data, the mortality rate of affected PD patients seems to be somewhere between 18 and 25% [31]. Whether these rates are associated to PD per se or are affected by advanced age, comorbidities and the potential frailty of the PD population remains to be investigated.

Interestingly, the all-cause mortality rate was found to be even higher (38.4%) in a recent UK-based multicenter study of 552 participants with a diagnosis of either PD or atypical parkinsonism who tested positive for SARS-CoV-2 while hospitalized [68]. Male sex, older age, frailty, dementia, the need for respiratory support and a lack of vaccination were identified as the most significant risk factors for increased mortality. Although this study does not refer specifically to COVID-19-related mortality, it highlights the role of other risk factors among PD/parkinsonism patients.

## 8. What Effect, If Any, Does COVID-19 Vaccination Have on People with Parkinson’s Disease?

There is a strong recommendation for PD patients to be vaccinated against COVID-19 unless specific contraindications exist [69]. COVID-19 vaccines are not expected to alter the effect of anti-parkinsonian medication, while their benefits and risks are believed to be the same for PD patients and their age-matched counterparts [70].

At present, there are limited data that parkinsonian symptoms may deteriorate following COVID-19 vaccination. More specifically, a survey of 14 PD patients reported a subjective worsening of their parkinsonian symptoms after COVID-19 vaccination [71]. Additionally, three PD patients, including a patient on LCIG treatment, developed troublesome dyskinesia post COVID-19 vaccination, which was successfully managed by lowering the total dopaminergic dose [72,73]. One PD patient on DBS reported an aggravation of both motor symptoms and NMSs after receiving the third booster dose of the COVID-19 vaccine, with the symptoms responding efficiently to modifications of the DBS settings [73]. Finally, two PD patients experienced deteriorations in their motor symptoms (rigidity, gait difficulties and tremor) after the first dose of the COVID-19 vaccination, followed by a spontaneous improvement [74].

## 9. Are There Any Treatment Options for Parkinson’s Patients with COVID-19?

An observational, retrospective multicenter study (*n* = 552) investigated the incidence and severity of COVID-19 infection in PD patients who were chronically prescribed amantadine compared to those who were not. No statistically significant differences were found between the two groups of patients, leading the authors to conclude that amantadine did not impair the risk of developing COVID-19 or the severity of the infection [33].

Vitamin D supplementation in PD patients was found to lower the risk of COVID-19 and COVID-19-related complications [75,76].

## 10. Pathophysiological Mechanisms

### 10.1. New Cases of Parkinsonism following COVID-19

Parkinsonism may develop following viral infections, including COVID-19, through immediate or delayed mechanisms, leading to para- or post-infectious forms of parkinsonism, respectively, although the two types of damage may overlap [77]. Regarding COVID-19-related parkinsonism, para-infectious cases are thought to occur within the first 15 days of contracting a SARS-CoV-2 infection due to the direct (the neuroinvasive potential of COVID-19) or indirect impairment of the nigrostriatal pathway (inflammatory, vascular and/or hypoxic injuries) [3]. On the other hand, post-infectious cases appear later and are attributed to COVID-19-triggered autoimmune reactions [3].

#### 10.1.1. The Neuroinvasive Potential of SARS-CoV-2

The neuroinvasive potential of COVID-19 remains controversial, with only a few autopsy studies detecting viral proteins or RNA in the brain or CSF of COVID-19 patients via immunostaining methods, none of them with new-onset parkinsonism [78]. More specifically, Stein at al. have not only confirmed the neuroinvasive potential of COVID-19 infection in their post-mortem case series describing 44 patients who died with COVID-19 but also showed that viral replication may take place in various tissues outside the respiratory tract and persist for months after developing COVID-19 [79]. Interestingly, and unlike other known models of coronavirus neurotropism, Emmi et al. detected viral proteins and genomic sequences of SARS-CoV-2 in the substantia nigra of patients who died in the acute phase of COVID-19 [80] and also within the vagal nuclei neurons in the medulla, suggesting a potential vagus-mediated tropism of SARS-CoV-2 in line with previous findings of Matschke et al. [81].

Several autopsy studies have identified an increased viral load in the olfactory mucosa, leading researchers to suggest the potential olfactory–transmucosal spread of SARS-CoV-2 in the CNS. However, these findings were not unanimous [80] and the virus was detected solely in the epithelium, not the olfactory neurons [82,83]. Since the pathological burden of α-synuclein in the olfactory bulb is thought to play a cardinal role in α-synucleinopathies, such as PD [84], and olfactory epithelial cells express the angiotensin-converting enzyme 2 (ACE2), the receptor assumed to mediate the entry of the COVID-19 virus into the cell [85], a significant amount of discussion has taken place considering the role of COVID-19 in the pathogenesis of PD [86,87]. Although olfactory structures and vagal nuclei represent areas of early PD pathology [88], such scenarios remain highly hypothetical and do not confirm direct viral neuronal damage, as hypoxic/ischemic insults and systemic inflammation may confound the observed neuropathological alteration in COVID-19. Larger autopsy studies, including individuals with histories of COVID-19 with various degrees of severity with or without persistent long-coronavirus symptoms, are required.

#### 10.1.2. COVID-19-Related Neuroinflammation

According to post-mortem findings, neuroinflammation, including lympho-monocytic infiltrations, microglial activation and microglial nodules, is pronounced in the context of COVID-19 [78]. SARS-CoV-2 infection is believed to trigger systematic inflammation and induce the release of cytokines release peripherally, even causing cytokine storm syndrome in severe cases of COVID-19 [89]. The activation of inflammatory mediators may lead to the disruption of the integrity of the blood–brain barrier (BBB), which may allow immune/inflammatory cells, cytokines or even the virus per se to infiltrate the CNS [90]. A neuroinflammatory cascade can be subsequently triggered, leading to increased oxidative stress and neuronal cell death [91]. Neuroinflammation has been suspected to promote neurodegeneration and contribute to the pathogenesis of PD, with midbrain dopamine neurons being particularly vulnerable to systemic inflammation due to their high energy requirements [92]. The activation of microglia and other indications of inflammatory changes have been confirmed in post-mortem brain studies [81,93].

Whether these neuroinflammatory processes follow a COVID-19-specific pattern or may be affected by patients’ comorbidities and/or hypoxic/ischemic insults remains to be clarified. Indeed, COVID-19 has been associated with a hypercoagulable state which is closely related to inflammation and thromboses, leading to arterial and venous infarcts [94].

#### 10.1.3. COVID-19-Related Immune Changes

Potential COVID-19-triggered autoimmune reactions are thought to involve molecular mimicry, bystander activation and viral persistence with or without epitope spreading [3]. This classification remains theoretical, and no autoantibodies have been detected in the CSF of patients with post-COVID-19 parkinsonism. However, as already mentioned, immunotherapy has been efficiently administered in some of the reported cases of parkinsonism, thus supporting an immune-mediated substrate.

### 10.2. New Cases of PD following COVID-19

Pathogens, including viruses, may be involved in the multicomplex pathogenesis of PD [20], with numerous promising theories arising over the past years (the dual-hit and multiple-hit hypotheses and the clustering of PD theory) [95,96,97]. Researchers argue that stressors, including viruses, may unmask underlying yet asymptomatic PD or precipitate a chain of events, resulting in the development of PD in selected, genetically susceptible individuals [98]. Becker et al. described two cases of PD with motor symptoms presenting shortly after COVID-19 [99]. After careful history taking, both patients exhibited NMSs which were compatible with a diagnosis of prodromal PD and long preceded the SARS-CoV-2 infection, leading authors to suggest that COVID-19 unmasked subjacent, pre-motor PD.

Beatman et al. suggested an antiviral role of α-synuclein [100]. More specifically, infections can precipitate the amplified neuronal expression of α-synuclein, which may also mediate interferon responses [101]. Viral proteins of COVID-19 were found to accelerate the upregulation and aggregation of α-syn, and their presence was associated with the formation of Lewy body pathology in vitro [102,103]. COVID-19 may, thus, predispose individuals to neurodegeneration and synucleinopathies, especially in the presence of inflammation, other stressors or genetic susceptibility. However, no robust evidence is available yet to support this scenario.

Outeiro et al. suggested that underlying inflammation precipitated by COVID-19 might accelerate biological aging disproportionally to chronological aging, causing an emergence of PD in younger age groups and offering an indirect link of COVID-19 to new PD cases [104]. Overall, COVID-19 infection may interfere with the homeostasis of numerous cellular structures, such as the mitochondria, the endoplasmic reticulum or the exosomes, whose dysfunction is thought to be involved in neuroinflammation and neurodegeneration, including the pathogenesis of PD [105].

An interesting hypothesis about exosomes, which are considered the primary regulators of intercellular communication, was recently reported. According to Mysiris et al. (2022), these extracellular vesicles, which can pass through the BBB due to their biocompatibility and bilayer lipid structure, may transfer the genetic material and proteins of SARS-CoV-2 into the CNS, shielding them from degradation [105]. However, focused testing is necessary to validate this theory.

### 10.3. Parkinson’s Patients: Do Symptoms of Parkinson’s Deteriorate during COVID-19?

The mechanisms mediating the aggravation of parkinsonian symptoms in PD patients who undergo an infection are not clear. Alterations in the regular schemes of anti-parkinsonian medications (e.g., due to hospitalization), along with impairment in pharmacodynamics/pharmacokinetics due to COVID-19 related symptoms (e.g., vomiting or diarrhea) or drug-to-drug interactions may explain, at least partly, these phenomena [106].

Various processes in the metabolism of dopamine can be impaired during systemic infections [107]. SARS-CoV-2 infection in particular may cause the downregulation of ACE2 receptors, which may in turn lead to the downregulation of the enzyme aromatic L-Dopa decarboxylase (DDC), with the latter being a crucial enzyme in the biosynthesis of dopamine [108]. Mpekoulis et al. found an inverse relationship between the RNA levels of SARS-CoV-2 and the expression of DDC in patients with COVID-19, implying that dopamine biosynthesis may be more impaired in COVID-19 patients with more severe symptoms [109]. Whether such a disruption may affect PD patients’ symptoms remains unclear. COVID-19-related stress may also play a role in the worsening of PD symptoms during an infection, although this is not confirmed [58]. Finally, the mechanisms underlying COVID-19-related damage to the CNS (e.g., neuro-invasion, neuroinflammation nad hypoxia) may take their toll on the clinical symptoms of PD, either by introducing novel symptoms or by worsening pre-existent parkinsonian features. Although the above mechanisms remain speculative and do not have a clear link to clinical manifestations of PD, they pave the way for future research in order to better understand the impact of infections, particularly SARS-CoV-2, on the pathophysiology of PD and to guide treatment options to mitigate or even reverse such effects.

The long-term follow-up of PD patients with a history of COVID-19, including assessments of various biomarkers (imaging, laboratory), may further elucidate subjacent pathophysiologic pathways and allow us to better understand the potential intertwined biological pathways of PD and COVID-19.

## 11. Conclusions

There is no doubt that the COVID-19 pandemic had an impact on PD patients whereby the latter became vulnerable to an overall worsening of their symptoms. Similar to other infections, including viruses, such negative consequences may be a direct effect of SARS-CoV-2 on impairing dopaminergic neurotransmission or an indirect result of the circumstances associated with an infection (e.g., hospitalization, fever, diarrhea and drug-to-drug interactions). Parameters related to the quarantine, such as social deprivation, loneliness and disruptions to medical care or exercise/physiotherapy, should also be considered. On the other hand, the relationship between COVID-19 and new-onset PD remains highly speculative, although the number of reported cases is growing, allowing us to make assumptions about the potential effect of SARS-CoV-2 on neurodegeneration. The time window of newly emerging cases of parkinsonism may serve as a criterion to roughly sort them as post- or para-infectious cases, suggesting a direct CNS insult due to the presumed neuroinvasive potential of SARS-CoV-2 or immune-mediated damage, respectively. As our knowledge derived from relevant cases broadens, such classifications may help us better understand the subjacent pathophysiology of these clinical presentations and properly address them therapeutically.

Reported variations in COVID-19-related parameters, including prevalence and outcomes, among different cohorts of PD patients should be interpreted with caution due to the heterogeneity of PD among the enrolled patients and the research methods used. Indeed, patients of different stages of PD or ages may have been variously affected by the sequelae of COVID-19. We should also bear in mind that data derived from the first wave of COVID-19 may be inherently different from data regarding the following years due to quarantine-related issues and a lack of vaccination.

As of 5 May 2023, the World Health Organization (WHO) lifted the “public health emergency of international concern” for COVID-19. However, as we are entering the long-term phase of COVID-19, longitudinal studies are imperative to map and explore how different aspects of PD are impacted after exposure to SARS-CoV-2 and other viruses and also to vaccines.

## Figures and Tables

**Table 1 biomedicines-11-02524-t001:** Summary of SARS-CoV-2 and cases of Parkinson’s disease.

Deterioration Features in People with Parkinson’s and COVID-19		References
**Motor performance overall**		[41,44,45,46]
OFF time reduction		[43]
Motor fluctuations		[42]
**Non-motor symptoms overall**		[41]
Cognition	COVID (+)	[4,43,44,48]
	COVID-19 status not specified	[42,49,50]
Mood/anxiety	COVID (+)	[41,44]
	COVID-19 status not specified	[41,44,47,49,51,52,53,54,55]
Sleep impairment	COVID (+)	[41]
	COVID-19 status not specified	[51,56]
Fatigue	COVID (+)	[41,43]
	COVID-19 status not specified	[44]
Pain	COVID (+)	[48]
	COVID-19 status not specified	[57]

PD: Parkinson’s disease; COVID (+): people who tested positive for a SARS-CoV-2 infection.

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
