# Peer review of "SARS-CoV-2 and Parkinson’s Disease: A Review of Where We Are Now"

_biomedicines, 2023, doi:10.3390/biomedicines11092524_

Round 1
Reviewer 1 Report
The review suggests that COVID-19 may have an impact on the onset and progression of PD. There are reports of new-onset movement disorders, including parkinsonism, appearing approximately two weeks after the beginning of COVID-19 and persisting for an average of 7.5 weeks in 22% of patients. Severe COVID-19 and encephalopathy were identified as potential risk factors. However, these findings are based on a limited number of studies and cases, and more research is needed to fully understand the potential impacts of COVID-19 on PD patients.
1. Provide a Clear Summary or Abstract: The paper could benefit from a clear and concise abstract or summary that provides an overview of the main points, findings, and conclusions. This would help readers quickly understand the key takeaways from the paper.
2. Expand on Limitations: While the authors have acknowledged some limitations, they could provide a more detailed discussion on how these limitations might impact the interpretation of their findings. This would give readers a better understanding of the context and potential biases in the study.
3. Provide More Context on Age Factor: The paper discusses the potential impact of COVID-19 on PD patients, but it does not clearly address the role of age as a risk factor. Given that PD is more common in older individuals, a more detailed discussion on how age might interact with COVID-19 to influence PD risk would be valuable.
4. Discuss Future Research Directions: The authors could provide more detailed suggestions for future research, including the types of studies that are needed to confirm their hypotheses, potential methodologies, and key research questions that remain unanswered.
5. Improve Organization and Flow: The paper could benefit from a more logical and clear organization of sections. This would improve the flow of the paper and make it easier for readers to follow the authors' line of reasoning.
6. Include More Recent Data: Given the rapidly evolving nature of COVID-19 research, the authors should ensure that they are including the most recent and relevant studies in their review. This could strengthen their arguments and ensure that their review is up-to-date.
The language used in the paper appears to be clear and precise, typical of scientific literature. It uses specific terminology related to the fields of neurology and virology, which might be challenging for a layperson to understand but is standard in academic and scientific discourse. However, there are a few points that could potentially benefit from further clarification:
1. The paper often refers to "PwP," which stands for "People with Parkinson's." This abbreviation is used frequently in the text and is a standard term in Parkinson's Disease research, but it might not be immediately clear to all readers.
2. The paper uses the term "Covid-19 positive PwP" to refer to people with Parkinson's who have also tested positive for COVID-19. This term is used to differentiate between Parkinson's patients who have and have not contracted COVID-19.
3. The paper discusses "motor" and "non-motor" symptoms. In the context of Parkinson's Disease, "motor" symptoms refer to physical symptoms such as tremors, stiffness, and problems with balance and coordination. "Non-motor" symptoms can include a wide range of issues such as cognitive changes, mood disorders, sleep problems, and autonomic dysfunction (issues with automatic body functions like blood pressure and digestion).
4. The term "OFF time" is used in the context of Parkinson's Disease to refer to periods when medication is not working well, and symptoms return or worsen.
Author Response
We thank the reviewer for the time taken to provide valuable feedback on our manuscript. We have worked together to hopefully address the report comments made and hope we have satisfied the reviewer with our changes.
Please see attached a breakdown of the reviewer 1 comments and our amendments outlined.

Reviewer 2 Report
The impact of SARS-CoV-2 on Parkinson's disease (PD) has been explored over the past three years. The connection appears complex, involving worsened PD symptoms due to the virus and potential degenerative changes. This review examines the relationship, encompassing pathophysiology, clinical effects, and vaccinations. The subject of this review is interesting and addresses pivotal factors. Before making a positive decision, some modifications are required.
Comments for authors
Comment 1: The abstract should effectively encapsulate the assessment of how COVID-19 influences Parkinson's disease, whether positively, negatively, or neutrally while offering crucial insights through recent statistics covering the main subject of this review.
Comment 2: For readers new to the subject, the authors' review and background might not provide adequate context for comprehending the study's importance. Incorporating additional information about Parkinson’s disease (PD) and its causative factors is advised. Notably, considering the emerging connection between neurodegenerative diseases like Alzheimer’s (AD) and Parkinson’s (PD) with factors such as microwave exposure, the authors should include relevant statements to enhance the scope of this review. Including recent articles could effectively address these aspects.
Article: Microwave Radiation and the Brain: Mechanisms, Current Status, and Future Prospects. International Journal of Molecular Sciences vol. 23 (2022). [https://doi.org/10.3390/ijms23169288].
Comment 3: How does the impairment of dopaminergic neurotransmission by SARS-CoV-2 contribute to the worsening of Parkinson's symptoms, and can this effect be reversed or mitigated? Explain in the manuscript.
Comment 4: What factors within the quarantine environment, such as social deprivation and disruptions in medical care, play a significant role in exacerbating Parkinson's symptoms, and can interventions alleviate these effects? It will be useful to discuss this in the manuscript.
Comment 5: In what ways might the Covid-19 sequelae affect patients with different stages of Parkinson's disease, and how can we predict and manage these effects based on individual disease characteristics? It will be useful for readers if authors are able to provide such information in this review.
Comment 6: What specific data points or biomarkers can be used to track the progression of Parkinson's disease in individuals exposed to Covid-19 and other viruses, and how can these markers guide clinical management? Is there any available literature on this, if yes include it in the manuscript.
The manuscript contains minor grammatical errors that require careful review and correction.
Author Response
We thank the reviewer for the time taken to provide valuable feedback on our manuscript. We have worked together to hopefully address the report comments made and hope we have satisfied the reviewer with our changes.
Please see attached a breakdown of the reviewer comments and our amendments outlined.

Round 2
Reviewer 1 Report
Authors addressed all issues raised by reviewer. I don't have further comment on this article.
Reviewer 2 Report
I recommend accepting the paper in its present form.